# NSAID use is not associated with Parkinson's disease incidence: A Norwegian Prescription Database study

Brage Brakedal [1,2], Charalampos Tzoulis[1,2], Ole-Bjørn Tysnes[1,2], Kristoffer Haugarvoll [1,2]*

1 Neuro-SysMed, Department of Neurology, Haukeland University Hospital, Bergen, Norway, 2 Department of Clinical Medicine, University of Bergen, Bergen, Norway

* haugarvoll@gmail.com

## Abstract

### Objective

Whether use of nonsteroidal anti-inflammatory drugs (NSAIDs) reduce the risk of incident Parkinson's disease (PD) remains unresolved. Here, we employed the Norwegian Prescription Database to examine whether NSAID use is associated with a lower incidence of PD.

### Methods

We compared the incidence of PD among users of NSAIDs in a population-based retrospective study using the Norwegian Prescription Database from 2004 to 2017. In total 7580 PD patients were identified using dopaminergic therapy over time as proxy for PD diagnosis. Analyses were performed with minimum 90 and 365 defined daily dose (DDD) NSAID exposure, respectively. Time-dependent Cox regression model and a binary logistic regression analysis with a 5-year lag until PD diagnosis were performed for all NSAIDs.

### Results

There was overall no decrease in incidence of PD among NSAID users compared to controls. Using a minimum of 90 or 365 DDD threshold of exposure produced similar results. Analysis of individual NSAIDs did not show difference in PD incidence compared to controls Age-specific incidence rates of PD were comparable to reported age-specific incidence rates in previous studies.

### Interpretation

Our findings provide no evidence that cumulative high exposure to NSAIDs affects the risk of developing PD.

**Data Availability Statement:** Data from the prescription registry can be acquired by applying to the Norwegian Prescription Registry. (www.reseptregisteret.no). Email: reseptregisteret@fhi.

no. Address: Folkehelseinstituttet, Avdeling for Helseregistre v/Reseptregisteret, Postboks 222 Skøyen, 0213 Oslo, Norway.

**Funding:** B.B: This work was supported by grants from the Western Norway Regional Health Authority (F11470). https://helse-vest.no/en The funder had no role in study design, data collection and analysis, decision to publish, or preparation of the manuscript.

**Competing interests:** The authors have declared that no competing interests exist.

**Abbreviations:** ASA, Acetylsalicylic Acid; ATC, Anatomical Therapeutic Code; CI, Confidence Interval; DDD, Defined Daily Dose; HR, Hazard ratio; NorPD, Norwegian Prescription Database; NSAID, Nonsteroidal anti-inflammatory drugs; PD, Parkinson's disease.

## Introduction

Parkinson's disease (PD) is the second most common neurodegenerative disorder [1]. While the etiology and pathogenesis of PD remain largely unknown, several processes have been implicated in its pathophysiology, including lysosomal and mitochondrial dysfunction, as well as neuroinflammation [2, 3]. Several studies have examined whether nonsteroidal anti-inflammatory drugs (NSAIDs) may decrease the risk of developing PD. Despite promising results from animal studies, the epidemiological evidence of NSAID-use with respect to PD risk is conflicting. Acetylsalicylic acid (ASA) has consistently not been associated with PD-risk [4–6]. A meta-analysis published in 2010 suggested that regular and long-term non-ASA NSAIDs use could have a protective effect, but the individual studies included in the meta-analysis showed conflicting results [6]. Another meta-analysis suggested that ibuprofen may be protective whereas a recent meta-analysis did not find that NSAIDs in general reduced the risk of PD [7–9]. It remains therefore unclear whether the use of non-ASA NSAID reduces the risk of PD. We collected all prescriptions of NSAIDs from 2004 to 2017 in the Norwegian Prescription database (NorPD) and assessed whether NSAID use is associated with a lower incidence of PD. Because PD is treated with dopaminergic drugs that are prescribed, we use dispensed dopaminergic drugs specific to PD as a proxy for PD diagnosis [10–14].

## Methods

### Ethical considerations

The data delivered by Norwegian Prescription Database (NorPD, www.reseptregisteret.no) was pseudo-anonymised. No consents were required as approved by NorPD and the Regional ethics committee. No additional records regarding the subjects were obtained.

The study was approved by the Norwegian Prescription Database (PDB 2417) and the Regional Committee for Medical and Health Research Ethics, Western Norway (REK 2016/1912).

### Material

Our study was based on the Norwegian Prescription Database (NorPD, www.reseptregisteret.no), an unselected, population-based registry of all drug prescriptions dispensed from Norwegian pharmacies to individual patients. Over the counter (OTC) drugs and drugs dispensed in institutions are not included. The NorPD comprises a complete record of every dispensing of prescribed medication from pharmacies since 01/01/2004 for the entire Norwegian population (5.1 million in 2013). NorPD has complete records of all deaths in Norway. The clinical indication for each prescription is registered in the form of either a diagnosis code from the International Classification of Diseases, 10th revision (ICD-10), and/or the International Classification of Primary Care, 2nd edition (ICPC-2), or a disease or disease-group specific reimbursement code. We included all acetylsalicylic acid (ASA), NSAID and anti-Parkinson (Anti-PD) drug prescriptions administered between 01/01/2004 and 31/12/2017. ASA drugs were identified by Anatomical Therapeutic Code (ATC) code B01AC06. NSIADs were identified by the ATC Codes; M01A*. All NSAID medications that were not orally administered were excluded. Dopaminergic PD medication is always reimbursed in Norway and is strictly prescription-controlled. PD incidence was defined by proxy from the use of levodopa (ATC Code: N04BA02, N04BA03), monoamine oxidase B inhibitors (ATC Code: N04BD01, N04BD02, N04BD03), or dopamine agonist (ATC Code: N04BC04, N04BC05, N04BC09) either alone or in combination, dispensed at least three consecutive times and at least 30 days apart. The time from the first to the last dopaminergic prescription had to be at least 180 days.

Prescriptions with non-PD reimbursement codes for the dopaminergic medications were excluded. Time of PD diagnosis was set to the first dispensed dopaminergic medication.

## Study design and selection of groups

All subjects had an observation period of 12 or more months. We defined 4 mutually exclusive groups (1–4): NSAID group(1), Sporadic NSAID group(2), NSAID+ASA group(3), and ASA group(4). Defined daily doses (DDD) that is the assumed average maintenance dose per day for a drug used for its main indication, was used to compute cumulative drug exposure (WHO Collaborating Centre for Drug Statistics Methodology, ATC classification index with DDDs, 2020. Oslo, Norway). We defined $\geq 90$ or $\geq 365$ cumulative DDD during the observation period as meaningful thresholds for NSAID exposure without introducing statistical or selection biases. NSAID group(1) were defined as being prescribed $\geq 90$ or $\geq 365$ cumulative DDD of NSAIDs and less than 90 cumulative DDD of ASA during the follow up. Sporadic NSAID group(2) were defined by being prescribed less than 90 or 365 cumulative DDD of NSAIDs and less than 90 cumulative DDD of ASA during the observation period. NSAID plus ASA group(3) were defined by being prescribed $\geq 90$ or $\geq 365$ cumulative DDD of NSAIDs and $\geq 90$ cumulative DDD of ASA during the follow up. ASA group(4) was defined by being prescribed $\geq 90$ cumulative DDD of ASA and $\leq 90$ cumulative DDD of NSAIDs prescriptions during follow up. Secondary analysis was likewise performed for diclofenac (ATC code: M01AB05 & M01AB55), ibuprofen (ATC code: M01AE01) and naproxen (ATC code: M01AE02 & M01AE52) where we required similar minimum of 90 or 365 of cumulative DDD thresholds of exposure. Similarly, the individual NSAIDs were group into those that had less than 90 cumulative DDD of ASA and more than 90 cumulative DDD of ASA exposure. For the time-dependent Cox regression analysis the follow up period started 12 months after the first dispensed prescription to ensure a minimum exposure time and follow up from the first prescription. Follow up continued to one of three endpoints; final observation time (31/12/2017), time of death or incidence of PD. We excluded subjects who were less 50 years old at the beginning of follow-up. Under the assumption that there is a potential latency period from exposure to outcome (PD incidence) we performed a 5-year lag analysis. For the 5-year lag analysis we pooled the prescriptions in 5 consecutive years and started the follow up after the 5-year lag and followed them until one of the three endpoints. A 5-year lag analysis would also eliminate any of immortal time bias from analysis. Subjects had to be classified to one of the four groups during the 5-year lag period. We stratified the groups to ensure that subjects included in one of the four groups during the 5-year lag would not change group during follow up.

## Statistical methods

In each analysis we compared the NSAID group(1) to the Sporadic NSAID group(2) and the NSAID plus ASA group(3) to the ASA group(4). This was to correct for ASA and because this created groups that had similar age and sex distribution. The individual NSAIDs, diclofenac, ibuprofen and naproxen, were compared against the Sporadic NSAID group (2) or ASA group (4) in all analysis depending whether they had less than or more than 90 DDD of ASA exposure respectively. To examine the incidence of PD between the groups we performed a time-dependent Cox regression model to correct for immortal time bias [15]. Immortal time, the time until an individual reached 90 or 365 of cumulative DDD exposure, was classified as follow-up time for the control group. The event in the time-dependent Cox regression model was PD diagnosis and subjects who died during the observation period were censored. The time-dependent Cox regression analysis was adjusted using age (at endpoint) and sex as fixed

covariates. We estimated the hazard ratio (HR) and 95% confidence interval (CI) for the incidence of PD. The age and sex covariates had a significant contribution to the model and satisfied the proportional hazard assumption which was verified by examining the log(-log (survival)) curves.

For the 5-year lag analyses we performed a binary logistic regression model and included age (at endpoint) and sex as covariates. We performed a Hosmer-Lemeshow goodness-of-fit test. All data processing and analysis were performed using IBM SPSS Statistics for Windows, Version 25.0. (Armonk, NY: IBM Corp).

## Results

Demographics for the overall NSAID groups used for analysis are presented in Tables 1 and 2. Demographics for the specific NSAIDS groups (diclofenac, ibuprofen and naproxen) used for analysis are presented in S1 and S2 Tables. Every NSAID group were demographically fairly similar to their control group. The most common prescribed non-ASA NSAIDs in Norway were diclofenac, ibuprofen and naproxen. All NSAID groups had a similar skewed distribution toward shorter time for the time from the last dispensed NSAID prescription until an endpoint. The ratio of immortal time compared against the total follow up time was also fairly consistent across the groups used for the time-dependent Cox regression analysis (Table 1 and S1 Table). Age-adjusted incidence rates for the Sporadic NSAID group(2) and ASA group(3) are shown in Table 2. The NSAID group(1) compared to the Sporadic NSAID group(2) and the NSAID plus ASA group(3) group compared to the ASA group(4) had a similar Q-Q plot distribution for age in both the time-dependent Cox regression analysis and binary logistic 5-year lag analysis. The specific NSAIDs (diclofenac, ibuprofen and naproxen) also had similar Q-Q plot distributions for age in comparison to their respective control group. The ratio of the total time in study to immortal time (time from first prescription until reaching 90 or 365 cumulative DDD thresholds) was similar across the NSAID groups with respect to ASA (NSAID groups with less than 90 of cumulative DDD of ASA and NSAID groups with more than 90 of cumulative DDD of ASA) (Table 1 & S1 Table). The age-specific incidence of PD in the Sporadic NSAID group(2) and ASA group (4) was similar to that reported in literature (Table 3) [16].

The main results for the time-dependent Cox regression analysis and the logistic regression analysis are shown in Table 4 Overall, almost every time-dependent Cox regression analysis and binary logistic 5-year lag analysis showed no difference or increased incidence of PD when the NSAID group(1), NSAID plus ASA group(3) or the equivalent NSAID specific group (diclofenac, ibuprofen and naproxen) was compared to their respective control group for both the 90 DDD and 365 DDD cumulative NSAID threshold (Table 4). There is concordance between the time-dependent Cox regression analysis and the binary logistic 5-year lag regression analysis. There was also concordance between the analyses that used threshold of 90 and 365 DDD of cumulative NSAID exposure. The only analysis that found a trend toward lower incidence rate of PD was analysis of Naproxen. This was not, however consistent across the different analyses. Thus this likely indicates that there was variation in some of the analysis, this could be due to the fact that the Naproxen group was the smallest group included in our analyses. The hazard ratio for age and sex was similar across all analysis for both the timed-dependent Cox regression and binary logistic 5-year lag analysis.

## Discussion

In this retrospective study, based on the entire Norwegian population, we found no evidence of an association between NSAID use and a decreased incidence of PD. Overall we observed

**Table 1. Demographics and descriptive statistics for Norwegian Prescription Database groups used in time-dependent Cox regression analysis.**

| Demographics 90DDD[a] Threshold: | NSAID(1)[b] | Sporadic(2)[c] | NSAID+ASA(3)[d] | ASA(4)[e] |
|---|---|---|---|---|
| Total Number: | 297.707 | 404.875 | 211.943 | 239.389 |
| Sex (Male %): | 37.4% | 46.8% | 46.0% | 55.7% |
| Follow-up time mean months (SD) | 78.4 (47.9) | 80.4 (43,9) | 86.9 (47.0) | 86.7 (43.2) |
| Total observation time (years)[f] | 2.683.756 | 2.713.360 | 1.980.983 | 1.730.596 |
| Total Immortal time(years)[g] | 738.825 | | 446.623 | |
| Mean age (SD) | 70.6 (9.2) | 68.7 (9.7) | 75.9 (9.5) | 76.0 (10.1) |
| Age 50–55 (%) | 2.867 (1%) | 13.578 (3%) | 319 (0.1%) | 1.119 (0.4%) |
| Age 55–60 (%) | 18.142 (6%) | 51.454 (13%) | 3299 (2%) | 7.498 (3%) |
| Age 60–65 (%) | 63.302 (21%) | 91.021 (22%) | 20.998 (10%) | 24.111 (10%) |
| Age 65–70 (%) | 72.863 (24%) | 85.557 (21%) | 36.891 (12%) | 36.909 (16%) |
| Age 70–75 (%) | 55.992 (19%) | 66.274 (16)% | 41.285 (17%) | 44.042 (18%) |
| Age 75–80 (%) | 32.923 (11%) | 37.162 (9%) | 33.911 (16%) | 36.268 (15%) |
| Age > 80 (%) | 51.618 (17%) | 59.829 (15%) | 75240 (36%) | 89.442 (37%) |
| Median cumulative NSAID DDD exposure[h] | 256 | 30 | 280 | 30 |
| Deaths (%) | 13.7% | 12.6% | 24.0% | 26.2% |
| Parkinson's disease, Incidence number | 1.760 | 2.354 | 1.467 | 1.999 |
| Demographics 365DDD[i] Threshold | NSAID(1)[a] | NSAID+ASA(3)[c] | | |
| Total Number | 113.791 | 86.896 | | |
| Sex (Male %): | 32.6% | 41.2% | | |
| Follow-up time mean months (SD) | 76.1 (48.1) | 84.3 (47.2) | | |
| Total observation time (years)[f] | 1.127.242 | 875.701 | | |
| Total Immortal time(years)[g] | 405.196 | 265.054 | | |
| Mean age (SD) | 71.9 (9.1) | 76.3 (10.2) | | |
| Age 50–55 (%)[e] | 458 (0.4%) | 65 (0%) | | |
| Age 55–60 (%) | 3.973 (4%) | 811 (1%) | | |
| Age 60–65 (%) | 20.826 (18%) | 7.441 (9%) | | |
| Age 65–70 (%) | 28.184 (25%) | 15.036 (17%) | | |
| Age 70–75 (%) | 23.192 (20%) | 17.519 (20%) | | |
| Age 75–80 (%) | 14.156 (12%) | 14.624 (17%) | | |
| Age>80 (%) | 23.002 (20%) | 31.400 (36%) | | |
| Median cumulative NSAID DDD exposure[d] | 801 | 803 | | |
| Deaths (%) | 3.7% | 4.3% | | |
| PD Incidence number | 714 | 606 | | |

[a] Minimum of 90 defined daily dose (DDD) exposure during follow up.

[b] NSAID group(1).

[c] Sporadic NSAID group(2).

[d] NSAID+ASA group(3).

[e] ASA group(4).

[f] total time for all subjects from first NSAID prescription until endpoint in years.

[g] total immortal time is the total time for all subjects until they reached the 90 or 365 cumulative DDD threshold.

[h] median of cumulative defined daily dose NSAID exposure during follow up.

[i] Minimum of 365 defined daily dose exposure during follow up.

no decrease in PD incidence in any of the analyses whether it was for NSAID exposure in general or for diclofenac, ibuprofen or naproxen in particular. Higher cumulative exposure, ≥365 DDD of NSAIDs, produced similar results to when we used ≥90 DDD of cumulative use of

**Table 2. Demographics and descriptive statistics for Norwegian Prescription Database groups used in binary logistic 5 year-lag analysis with 90 DDD[a] threshold.**

| Demographics 90DDD[a] Threshold: | NSAID(1)[b] | Sporadic(2)[c] | NSAID+ASA(3)[d] | ASA(4)[e] |
|---|---|---|---|---|
| Total Number: | 172.887 | 296.602 | 98.203 | 147.730 |
| Sex (Male %): | 35% | 45.7% | 45% | 55.9% |
| Mean age (SD) | 71.8 (9.3) | 69.7 (9.0) | 77.7 (9.6) | 76.8 (9.9) |
| Age 55–60 (%) | 8.283 (5%) | 27.079 (9%) | 1.311 (1%) | 3.458 (2%) |
| Age 60–65 (%) | 32.970 (19%) | 71.337 (24%) | 7.236 (7%) | 13.406 (9%) |
| Age 65–70 (%) | 41.705 (24%) | 70.269 (24%) | 13.856 (14%) | 21.907 (15%) |
| Age 70–75 (%) | 34.276 (20%) | 54.157 (18)% | 17.524 (18%) | 27.272 (18%) |
| Age 75–80 (%) | 20.576 (12%) | 29.027 (10%) | 15.820 (16%) | 22.739 (15%) |
| Age > 80 (%) | 35.077 (20%) | 44.733 (15%) | 42.456 (43%) | 58.948 (40%) |
| Median cumulative NSAID DDD exposure[d] | 398 | 30 | 362 | 30 |
| Deaths (%) | 14% | 9.7% | 28.4% | 23.8% |
| PD Incidence number | 899 | 1.181 | 655 | 910 |
| Demographics 365DDD[g] Threshold | NSAID(1)[b] | NSAID+ASA(3)[d] | | |
| Total Number | 56.396 | 34.286 | | |
| Sex (Male %): | 30% | 40.0% | | |
| Follow-up time mean months (SD) | 111.0 (37.2) | 107.7 (37.7) | | |
| Mean age (SD) | 73.5 (9.4) | 78.6 (9.4) | | |
| Age 55–60 (%) | 1590 (3%) | 289 (1%) | | |
| Age 60–65 (%) | 8.053 (14%) | 1.963 (6%) | | |
| Age 65–70 (%) | 12.829 (23%) | 4.385 (13%) | | |
| Age 70–75 (%) | 11.969 (21%) | 6.097 (18%) | | |
| Age 75–80 (%) | 7.615 (14%) | 5.679 (16%) | | |
| Age>80 (%) | 14.340 (25%) | 15.873 (46%) | | |
| Median cumulative NSAID DDD exposure [f] | 1250 | 1035 | | |
| Deaths (%) | 18.9% | 33.7% | | |
| PD Incidence number | 355 | 249 | | |

[a] Minimum of 90 defined daily dose (DDD) exposure during follow up.

[b] NSAID group(1).

[c] Sporadic NSAID group(2).

[d] NSAID+ASA group(3).

[e] ASA group(4).

[f] median of cumulative defined daily dose NSAID exposure during follow up.

[g] Minimum of 365 defined daily dose exposure during follow up.

**Table 3. Age-adjusted incidence rates for Norwegian Prescription Database.**

| Age Group | No. Of PD cases | Person-years | Incidence rate |
|---|---|---|---|
| 50–59 | 306 | 266.924 | 11.4 |
| 60–69 | 1.337 | 1.764.281 | 76.7 |
| 70–79 | 1.701 | 1.400.369 | 121.4 |
| >80 | 1.009 | 1.012.382 | 99.6 |

Age-Adjusted incidence rate for the Sporadic NSAID group(2) and ASA group(4). Incidence rate is number of Parkinson's disease incidences per 100.000 Person-years.

**Table 4. Results from time-dependent Cox regression analysis and binary logistic regression analysis.**

| | DDD[a] threshold | Time-dependent Cox Regression analysis | | | Binary Logistic Regression 5-year lag analysis | | |
|---|---|---|---|---|---|---|---|
| | | p-value | HR[b] | CI[c] | p-value | Exp(B)[d] | CI |
| **NSAID: A1[e]** | 90 | <0.001 | 1.44 | 1.35–1.53 | <0.001 | 1.30 | 1.19–1.42 |
| **NSAID: A2[f]** | 90 | 0.18 | 1.05 | 0.98–1.23 | 0.012 | 1.14 | 1.03–1.26 |
| **NSAID: A1** | 365 | <0.001 | 1.48 | 1.36–1.62 | <0.001 | 1.53 | 1.35–1.72 |
| **NSAID: A2** | 365 | 0.55 | 1.03 | 0.94–1.54 | 0.002 | 1.25 | 1.08–1.44 |
| **Diclofenac: A1[g]** | 90 | <0.001 | 1.21 | 1.11–1.31 | 0.008 | 1.17 | 1.04–1.31 |
| **Diclofenac: A2[h]** | 90 | 0.41 | 0.96 | 0.88–1.06 | 0.90 | 1.01 | 0.88–1.16 |
| **Diclofenac: A1** | 365 | <0.001 | 1.28 | 1.11–1.47 | 0.008 | 1.29 | 1.06–1.57 |
| **Diclofenac: A2** | 365 | 0.62 | 0.96 | 0.83–1.15 | 0.20 | 1.15 | 0.92–1.44 |
| **Ibuprofen: A1[i]** | 90 | 0.051 | 1.11 | 0.99–1.24 | 0.60 | 1.04 | 0.89–1.22 |
| **Ibuprofen: A2[j]** | 90 | 0.037 | 0.88 | 0.77–0.93 | 0.73 | 1.03 | 0.87–1.22 |
| **Ibuprofen: A1** | 365 | 0.11 | 1.15 | 0.97–1.37 | 0.16 | 1.18 | 0.93–1.51 |
| **Ibuprofen: A2** | 365 | 0.91 | 1.01 | 0.84–1.21 | 0.42 | 1.06 | 0.80–1.41 |
| **Naproxen: A1[k]** | 90 | 0.06 | 0.86 | 0.77–1.01 | 0.72 | 0.97 | 0.81–1.15 |
| **Naproxen: A2[l]** | 90 | 0.007 | 0.83 | 0.73–0.95 | <0.001 | 0.68 | 0.55–0.84 |
| **Naproxen: A1** | 365 | 0.97 | 0.97 | 0.82–1.22 | 0.27 | 1.17 | 0.89–1.53 |
| **Naproxen: A2** | 365 | 0.02 | 0.76 | 0.61–0.95 | 0.18 | 0.80 | 0.57–1.11 |

A1 is NSAID(or specific NSAID) compared against sporadic NSAID group(3). A2 is NSAID compared against ASA group(4).

[a]) Defined Daily Dose (DDD), minimum threshold for NSAID exposure in analysis.

[b]) Hazard ratio of group comparison.

[c]) Confidence Interval.

[d]) exponentiation of the B coefficient; odds ratio of group analysis.

[e]) NSAID group(1) compared against Sporadic NSAID group(2).

[f]) NSAID + ASA group (3) compared against ASA group(4).

[g]) Diclofenac group compared against sporadic NSAID group(3).

[h]) Diclofenac + ASA group (>90 DDD of ASA exposure) compared against ASA group(4).

[i]) Ibuprofen group compared against sporadic NSAID group(3).

[j]) Ibuprofen + ASA group (>90 DDD of ASA exposure) compared against ASA group(4).

[k]) Naproxen group compared against Sporadic NSAID group.

[l]) Naproxen + ASA group (>90 DDD of ASA exposure) compared against ASA group(4).

NSAIDs as the threshold. All groups compared in the analyses were consistently similar. The overall data support that NSAID does not associate with PD incidence, however we did observe that the general NSAID group(1) was associated with significant higher PD incidence compared to the sporadic NSAID group(2). In a similar analysis where we examined NSAID users who also had a significant ASA exposure against those who only used ASA was however not significant. It is therefore likely that there could be an underlying factor that influenced these results rather than NSAIDs themselves being associated with increased risk of PD. We found in some of the time-dependent Cox regression analyses that the hazard ratio (HR) could be slightly higher than the odds ratio of the logistic regression. An increased HR in the timed-dependent Cox regression analysis could be due to the correction of immortal time. Immortal time refers to the follow-up time during which the outcome could not have occurred. In our analysis this corresponds to the time until a subject reached the ≥90 or ≥365 DDD thresholds from the first prescription [17]. During the time a subject used NSAID until they reached the ≥90 or ≥365 DDD thresholds, they would be artificially protected against reaching one of the

study endpoints, hence this time period is referred to as "immortal". If one does not correct for the immortal time bias, this could give an increased survival advantage. One way to correct for immortal time is to classify the time before exposure as unexposed and as exposed thereafter [15]. Time-dependent Cox regression analysis is a good method to control for immortal time in the exposed group [17, 18]. Excluding immortal time could lead to an overestimation of the HR among non-users of NSAIDs, whereas adding immortal time to non-users could inflate the HR among the users [19]. When we performed the 5 year-lag logistic regression analysis, we found similar results, i.e. NSAID users did not have significant higher PD incidence compared to the NSAID non-users. We performed a stratified analysis on the three most prevalent used NSAIDs; diclofenac, ibuprofen and naproxen. Earlier studies have reported mixed results with respect to ibuprofen use as neuroprotective against PD incidence [5, 7, 20, 21]. Our results for ibuprofen were similar to the overall NSAID result where we did not detect any decreased incidence of PD among ibuprofen users. Likewise we found no overall evidence that ibuprofen or naproxen lowers PD incidence risk.

The main strength of this study is that it includes the entire population of Norway and every NSAID and PD drug prescribed over a period of 14 years. The very large numbers of subjects would reduce the risk of uneven distribution of age and sex, the two strongest risk factors for PD, in the analyzed samples. We report age-adjusted incidence rates that are very similar to the age-adjusted incidence rates reported previously reported [16, 22]. This suggests that the PD incidence reported here are representative of the true PD incidence in the Norwegian population.

A potential weakness to this study is the lack of possible confounding variables such as exposure (e.g. smoking) and medical history in the NorPD. However, studies have shown similar habits between NSAID users and nonusers with respect to smoking, caffeine and alcohol consumption [23]. Multiple other covariates and comorbidities have been included in earlier studies, but none or few have been found to be of significance, it is therefore unlikely that there is any major missing stratification that is biasing the results [24, 25]. Moreover, because of the very large number of subjects in each of the groups we expect that other comorbidity, use of other medication and environmental exposure will be distributed fairly evenly between the groups. Hence, the lack of comorbidity covariates is unlikely to significantly affect our risk estimates. An increased regular use of NSAID use could however be associated with chronic illnesses such as rheumatoid arthritis and chronic pain conditions. A few studies has found that rheumatoid arthritis is possibly associated with lower PD incidence, this however should produce a bias in favor of NSAID use being associated with reduced incidence of PD [26, 27]. Another weakness of this study is that it does not have data for over the counter drug (OTC) use among the subjects. OTC drugs such as ibuprofen are however only sold in lower dosage strength and small packages and is used mainly sporadically. In Norway paracetamol (ATC Code N02BE01) is the most commonly sold OTC analgetic drug. Diclofenac and Naproxen are not OTC drugs and require prescription. Though it is likely that average cumulative DDD could be a little higher for the Sporadic NSAID group or the ASA group it is unlikely to create a significant bias.

The method to identify Parkinson's Disease by using dopaminergic drugs as proxy for Parkinson disease is highly specific since dopaminergic drugs are coupled with a Parkinson's Disease reimbursement code and the combination of MAO-B, Levodopa and Dopamine Agonists is very specific to Parkinson Disease [10–12, 24, 28]. It is however difficult in the early stages of parkinsonism to differentiate PD from atypical parkinsonism as atypical parkinsonism could initially be treated with dopaminergic drugs, as part of the diagnostic work-up. The incidence of the most common causes of atypical parkinsonism is however very low compared to PD. The most common atypical parkinsonism that are usually initially treated with dopaminergic

drugs is multiple system atrophy (incidence of 0.8 per 100.000 year), progressive supranuclear palsy (incidence of 0.9 per 100.000 year) and corticobasal degeneration (0.2 per 100.000 year) and are thus very rare in comparison to PD [29]. Other causes to secondary parkinsonism such as drug-induced parkinsonism and vascular parkinsonism are unlikely to be treated with dopaminergic drugs and thus identified in this study as PD. It is however possible that we report a slight overestimate of PD incidence, but because of the large groups analyzed and high number of PD incidences in each group, it is unlikely that this should significantly impact the results.

In summary, this population-based retrospective study found no association between cumulative NSAID use and decreased incidence of PD. This was observed for NSAIDs in general and for diclofenac, ibuprofen or naproxen in particular.

## Supporting information

**S1 Table. Demographics and descriptive statistics for Norwegian Prescription registry groups used in time-dependent Cox regression analysis with 90DDD[a] threshold.** (DOCX)

**S2 Table. Demographics and descriptive statistics for Norwegian Prescription registry groups used in binary logistic 5-year lag regression analysis with 90DDD[a] threshold.** (DOCX)

## Author Contributions

**Conceptualization:** Brage Brakedal, Kristoffer Haugarvoll.

**Formal analysis:** Brage Brakedal.

**Funding acquisition:** Kristoffer Haugarvoll.

**Investigation:** Brage Brakedal.

**Methodology:** Brage Brakedal, Charalampos Tzoulis, Kristoffer Haugarvoll.

**Supervision:** Charalampos Tzoulis, Ole-Bjørn Tysnes, Kristoffer Haugarvoll.

**Writing – original draft:** Brage Brakedal, Kristoffer Haugarvoll.

**Writing – review & editing:** Brage Brakedal, Charalampos Tzoulis, Ole-Bjørn Tysnes, Kristoffer Haugarvoll.

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
