## [Decision Letter · Decision Letter 0]

14 Apr 2021

PONE-D-21-06959

NSAID use is not associated with Parkinson’s Disease incidence: A Norwegian Prescription Registry study

PLOS ONE

Dear Dr. Haugarvoll,

Thank you for submitting your manuscript to PLOS ONE. After careful consideration, we feel that it has merit but does not fully meet PLOS ONE’s publication criteria as it currently stands. Therefore, we invite you to submit a revised version of the manuscript that addresses the points raised during the review process.

I kindly suggest Authors to take into account the comments raised by both Reviewers.

We look forward to receiving your revised manuscript.

Kind regards,

Claudio Liguori

Academic Editor

PLOS ONE

Journal Requirements:

2. Please provide additional details regarding participant consent. In the ethics statement in the Methods and online submission information, please ensure that you have specified (i) whether consent was informed and (ii) what type you obtained (for instance, written or verbal, and if verbal, how it was documented and witnessed). If your study included minors, state whether you obtained consent from parents or guardians. If the need for consent was waived by the ethics committee, please include this information.

5. Please amend your list of authors on the manuscript to ensure that each author is linked to an affiliation. Authors’ affiliations should reflect the institution where the work was done (if authors moved subsequently, you can also list the new affiliation stating “current affiliation:….” as necessary).

Reviewers' comments:

Reviewer's Responses to Questions

**Comments to the Author**

1. Is the manuscript technically sound, and do the data support the conclusions?

Reviewer #1: Yes

Reviewer #2: Partly

2. Has the statistical analysis been performed appropriately and rigorously? 

Reviewer #1: Yes

Reviewer #2: Yes

3. Have the authors made all data underlying the findings in their manuscript fully available?

Reviewer #1: Yes

Reviewer #2: Yes

4. Is the manuscript presented in an intelligible fashion and written in standard English?

Reviewer #1: Yes

Reviewer #2: Yes

5. Review Comments to the Author

Reviewer #1: Review of the manuscript NSAID use is not associated with Parkinson’s Disease incidence: A Norwegian Prescription Registry study by Brakedal et al.

The paper exploits the Norwegian Prescription Database to investgate the incidence of Parkinson’ s disease (PD) among users of nonsteroidal anti-inflammatory drugs (NSAIDs). They find that there is no association between the use of NSAIDs and the risk of developing PD.

The idea of investigating the neuroprotective role of NSAIDs in PD is not novel, as several studies on this topic are published. There are some discrepancies on the effect of NSAIDs, but two quite recent meta-analyses concluded that the use of NSAIDs is not associated with the risk of PD (Ren L et al. Medicine (Baltimore) 2018; Poly TN Eur J Clin Pharmacol 2019). The former paper is not cited in the manuscript, but it should, as it examines the effect of subgroups of NSAIDs, as well as the impact of cumulative use of NSAIDs.

The strength of this study is that the Norwegian Prescription Database allows investigation of the entire population of Norway, making it one of the largest epidemiological study on the association between NSAIDs and PD.

I have some concerns that should be addressed before publication:

1. I wonder about the “cut off” of 90 defined daily doses (DDD) of NSAIDs between group 1 and 2 and the length of the observation period.

A) The authors say that all subjects had an observation period of 12 or more months. The observation time is from 01/01/04 to 31/12/17. Would this mean that a person in group 1) could have used an average dose of NSAIDs of 0.0016 DDD/day (90 DDD over 156 months), while a person in group 2) could have used an average dose of 0.24 DDD/day (89 DDD over 12 months)? Wouldn’t this make the person in group 2) more exposed than the person in group 1)?

B) What is the thinking behind the “cut off” of 90 DDD of NSAIDs? Will this make the persons in group 2) sufficiently “unexposed”, so that their NSAID use will not protect dopamine neurons from dying? Opposite, will the persons using more than 90 DDD over the observation time be sufficiently exposed, so that their use of NSAIDs will confer neuroprotection?

To elucidate the NSAID exposure in group 2) the authors could give data on the average cumulative NSAID dose before each person developed PD? This could be compared to the cumulative NSAID doses in persons in group 2) not developing PD.

C) Given that NSAIDs are seldom taken on a daily basis over a long period of time, but often in short periods when patients suffers from pain, I wonder how large was the gap in time between the last dose of NSAID and the development of PD. If this gap is very long (several months/years) do the authors believe that the NSAID use could be associated with development of PD? I would think that this could influence the chances of observing an association between the use NSAIDs and PD development.

As mentioned above I wonder about the comparison between group 1) and 2); is group 2) a control? Would it not be better to compare PD incidence in the NSAID exposed groups to the PD incidence in the Norwegian population not taking any NSAIDs?

3. Even though recent meta-analyses suggest that NSAIDs do not protect dopamine neurons from dying in PD, there are some discrepancies as described in the MS, in particular related to ibuprofen. Thus, would it not increase the validity of the author’s conclusion if they examined the relation between the use of ibuprofen and PD incidence separately? I also think it would of interest to the readers if the authors could present the main subtype of NSAIDs used by the persons in the study (also see below).

2. I think the manuscript is a bit short and technically written. I suppose the authors want to communicate with the general population of neurologists/neuroscientists. Some of the epidemiological terms therefore need to be described and explained.

A. Defined daily dose (DDD) should be explained. The authors should also refer to a site where the readers could find information about the dose in mg/g, which corresponds to 1 DDD (e.g. WHO Collaborating Centre for Drug Statistics Methodology, ATC classiﬁcation index with DDDs, 2020. Oslo, Norway), or the authors could make a table in which the DDD of the main NSAIDs investigated in the study is given.

B. Immortal time bias should be explained. The discussion of the impact of correction for immortal time bias in the Discussion is quite difficult to follow for generalists and should be rewritten to make the points more clear.

C. In Methods, p.6, l. 18-20 the authors say that “for the overall analysis the observation period started 12 months after the first dispensed prescription..”. The authors should explain why this was done. They should explain how only incident PD cases were included in the study.

D. The authors should explain the rationale for doing “the 5-year lag analysis”.

E. In Methods, p. 7, l. 11-13 the authors say that “the age and sex covariates had a significant contribution to the model and satisfied the proportional hazard assumption which was verified by examining the DFBETA residuals against time and log(-log[PD incident]) versus log(time) curved ”. The meaning of this is unclear and DFBETA is not defined.

3. Some co-morbidities, for which NSAIDs could be subscribed, may be related to PD development. Would the authors consider to correct for such co-morbidities? For instance inflammatory bowel disease seems to be associated with increased risk of PD (e.g. Villumsen M Gut 2019; Weimers P Inflamm Bowel Dis 2019). The same is true for diabetes ( e.g. Yue X Medicine 2016). If this is the case, could it mask the observed effect of NSAIDs on PD incidence in this study? Rheumatoid arthritis (RA) is probably related to a decreased risk of PD (Li C NPJ Parkinsons Dis 2021) and would not explain the results, unless the number of persons with RA was larger in the “control” groups in the study.

4. Minor

In Methods, p. 5, l. 23 the authors say that NSIADs were identified by the following ATC Codes; M01A* and S01BC*. The latter is eye preparations. The authors should clarify.

One finding was that there was a 17% higher hazard ratio for PD among NSAID users compared to those who did not use NSAID (although not reproduced in another regression model). Could it be that the effect of NSAIDs in PD cases, who have progressed into the motor phase, is to inhibit clearance of increased level of extracellular alpha-synuclein by microglia, thereby worsening the PD course?

In Introduction, p. 4, l. 14 the references cited (8 and 9) must be wrong.

In Discussion, p.11, l. 7-8: how could “the minimal stratification” both be a strength and a weakness?

Reviewer #2: Neuroinflammation is an increasingly important pathophysiological topic and, perhaps, a future therapeutic challenge in Parkinson's disease (PD).

In this study, Authors try to investigate indirectly neuroinflammation in PD, through a drug prescription registry study.

I appreciated the Authors' purpose and study design, in particular the choice to use group analysis based on the division by mean age (which exclude the confounding factor of co-morbidities).

Appropriate methods and analysis, well-argued discussion. Moreover, it is a well written article.

However I encountered some limitations.

1. I do not completely agree with the basic idea of assimilating neuroinflammation and systemic inflammation; consequently a study on NSAIDs may be too simplistic.

2. I would also look at the other side of the coin: not only the use of NSAIDs to reduce inflammation, but also the possibility that those who use NSAIDs probably start from a more pronounced inflammatory condition.

So, I ask to answer these main questions and to modify the discussion

Finally, I have just one minor concern, maybe my limitation, because I did not fully understand:

1. Does the Norwegian Prescribing Database take into account the whole population? In other words, were people who did not receive any prescriptions in their life (until the authors' research) also included in the Sporadic group?

---

## [Author Response · Author response to Decision Letter 0]

7 Jul 2021

Please see attached "Response to Reviewers" for comments and reply.

---

## [Decision Letter · Decision Letter 1]

11 Aug 2021

NSAID use is not associated with Parkinson’s Disease incidence: A Norwegian Prescription Database study

PONE-D-21-06959R1

Dear Dr. Haugarvoll,

We’re pleased to inform you that your manuscript has been judged scientifically suitable for publication and will be formally accepted for publication once it meets all outstanding technical requirements.

Kind regards,

Claudio Liguori

Academic Editor

PLOS ONE

Additional Editor Comments (optional):

Reviewers' comments:

Reviewer's Responses to Questions

**Comments to the Author**

1. If the authors have adequately addressed your comments raised in a previous round of review and you feel that this manuscript is now acceptable for publication, you may indicate that here to bypass the “Comments to the Author” section, enter your conflict of interest statement in the “Confidential to Editor” section, and submit your "Accept" recommendation.

Reviewer #1: All comments have been addressed

Reviewer #2: All comments have been addressed

2. Is the manuscript technically sound, and do the data support the conclusions?

Reviewer #1: Yes

Reviewer #2: Yes

3. Has the statistical analysis been performed appropriately and rigorously? 

Reviewer #1: Yes

Reviewer #2: Yes

4. Have the authors made all data underlying the findings in their manuscript fully available?

Reviewer #1: Yes

Reviewer #2: Yes

5. Is the manuscript presented in an intelligible fashion and written in standard English?

Reviewer #1: Yes

Reviewer #2: Yes

6. Review Comments to the Author

Reviewer #1: The authors have adequately answered my comments point by point. I have no further comments and the manuscript is ready for publication.

Reviewer #2: Although animal studies appear to show a reduction in the risk of developing Parkinson’s Disease (PD) using anti-inflammatory drugs (NSAIDs), epidemiological evidence on the use of NSAIDs in humans with respect to the risk of PD remains conflicting. Several studies examined the possible link between NSAID use and the risk of PD without a definitive indication.

In this retrospective study Authors employed the Norwegian Prescription database from 2004 to 2017 to examine whether NSAID may modify the risk of developing PD.

I appreciated Authors’ purpose and study design.

Clear rationale, appropriate methods and analysis, well-argued discussion: it is now clear how the use of a large prescription database (5.1 million in 2013) allows to make inferences on an entire national population and also allows to avoid diagnostic biases.

The major finding of this study is that there is no evidence that cumulative high exposure to NSAIDs affects the risk of developing PD.

Moreover, it is a well written article.

---

## [Editor Report · Acceptance letter]

27 Aug 2021

PONE-D-21-06959R1 

NSAID use is not associated with Parkinson’s Disease incidence: A Norwegian Prescription Database study 

Dear Dr. Haugarvoll:

I'm pleased to inform you that your manuscript has been deemed suitable for publication in PLOS ONE. Congratulations! Your manuscript is now with our production department. 

Kind regards, 

on behalf of

Dr. Claudio Liguori 

Academic Editor

PLOS ONE